# LncRNAs in Non-Small-Cell Lung Cancer

**DOI:** 10.3390/ncrna6030025

**Published:** 2020-06-30

**Authors:** Lucy Ginn, Lei Shi, Manuela La Montagna, Michela Garofalo

**Affiliations:** Transcriptional Networks in Lung Cancer Group, Cancer Research UK Manchester Institute, University of Manchester, Alderley Park, Manchester SK10 4TG, UK; lucy.ginn@postgrad.manchester.ac.uk (L.G.); lei.shi@manchester.ac.uk (L.S.); manuela.lamontagna@postgrad.manchester.ac.uk (M.L.M.)

**Keywords:** non-small-cell lung cancer, long non-coding RNAs

## Abstract

Lung cancer is associated with a high mortality, with around 1.8 million deaths worldwide in 2018. Non-small-cell lung cancer (NSCLC) accounts for around 85% of cases and, despite improvement in the management of NSCLC, most patients are diagnosed at advanced stage and the five-year survival remains around 15%. This highlights a need to identify novel ways to treat the disease to reduce the burden of NSCLC. Long non-coding RNAs (lncRNAs) are non-coding RNA molecules longer than 200 nucleotides in length which play important roles in gene expression and signaling pathways. Recently, lncRNAs were implicated in cancer, where their expression is dysregulated resulting in aberrant functions. LncRNAs were shown to function as both tumor suppressors and oncogenes in a variety of cancer types. Although there are a few well characterized lncRNAs in NSCLC, many lncRNAs remain un-characterized and their mechanisms of action largely unknown. LncRNAs have success as therapies in neurodegenerative diseases, and having a detailed understanding of their function in NSCLC may guide novel therapeutic approaches and strategies. This review discusses the role of lncRNAs in NSCLC tumorigenesis, highlighting their mechanisms of action and their clinical potential.

## 1. Introduction

Lung cancer is the leading cause of cancer related deaths worldwide, with 1.8 million deaths in men and women in 2018 [1]. Lung cancer is classified histologically into two main subtypes: small-cell lung cancer (SCLC) which accounts for around 15% of cases and non-small-cell lung cancer (NSCLC) which makes up about 85% of cases [2]. NSCLC is further grouped into adenocarcinoma (AC), squamous cell carcinoma (SCC), and large cell carcinoma (LCC), where ACs are the most prevalent accounting for around 40% of cases [3]. NSCLC is regarded as a heterogeneous disease, even within histological subgroups, owing to different molecular mechanisms driving lung tumorigenesis [4]. The development of targeted molecular therapies greatly improved patient response to the therapy, and their success demonstrates the benefit of treating biologically relevant alterations [3]. However, 40% of patients still present with stage IV lung cancer at diagnosis, with a five-year survival of 10–15%. This highlights a need to identify novel targets that may direct future therapies to reduce the burden of NSCLC [2,4].

In the last decade, the role of non-coding RNAs (ncRNA) as key regulators in a wide range of cellular processes became increasingly apparent [5]. NcRNAs are mainly grouped by size, and those smaller than 200 nucleotides (nt) are classified as small non-coding RNAs (sncRNAs), including the well-known housekeeping ncRNAs transfer RNAs (tRNAs), ribosomal RNAs (rRNAs), small nuclear RNAs (snRNAs), and small nucleolar RNAs (snoRNAs), as well as small interfering RNAs (siRNA), microRNAs (miRNAs), and piwi-interacting RNAs (piRNAs) [6,7]. Long non-coding RNAs (lncRNAs) are defined as non-protein coding RNA transcripts more than 200 nt in length [8,9]. Although lncRNAs are principally classified by length, they can also be categorized by their genomic location [10]. Based on their position relative to coding genes, lncRNAs can be classed as long intergenic RNAs (lincRNAs), which include enhancer RNAs (eRNAs) that are transcribed from distal enhancer regions, intronic lncRNAs, overlapping lncRNAs, sense lncRNAs, antisense lncRNAs, and bidirectional lncRNAs [11,12]. Through their interaction with chromatin, proteins, and RNA targets, lncRNAs can regulate gene expression and signaling pathways at the epigenetic, transcriptional, and post-transcriptional level, as reviewed in Schmitt et al. [9].

The processing of lncRNAs is similar to that of messenger RNAs (mRNAs) in that they are transcribed by RNA polymerase type II (RNAP2), showing similar methylation patterns across the gene body, and they undergo post-transcriptional modifications such as splicing, polyadenylation, and 5′ capping [10]. Although lncRNAs are less conserved than protein-coding genes, the promoter regions have high sequence conservation, suggesting the importance of lncRNA regulation [9,11]. The expression of lncRNAs is regulated at the transcriptional and epigenetic level and tightly controlled [13,14]. Although their expression is relatively low, lncRNAs are differentially expressed in tissues and have high tissue specificity [14]. Although tightly regulated in physiological tissues, lncRNAs are commonly dysregulated in disease, leading to aberrant expressions and functions [15,16].

Recently, lncRNAs were implicated in cancer, where they can function as oncogenes and tumor suppressors, and their dysregulation is associated with tumor cell growth, apoptosis, invasion, and metastasis [14,17]. Due to the function of lncRNAs in key pathophysiological pathways, they are gaining increasing attention as novel anti-cancer therapeutic targets [18]. Some lncRNAs are well characterized in NSCLC; however, the precise mechanisms of action for most lncRNAs remains largely unknown [14]. Here, we review the role of lncRNAs in NSCLC, focusing on their precise function in NSCLC tumorigenesis, their mechanism of action, and their clinical potential.

## 2. LncRNA Mechanisms of Action

The function of lncRNAs is largely reflected by their subcellular localization, where nuclear lncRNAs mainly function in transcriptional processes, whereas cytoplasmic lncRNAs tend to function post-transcriptionally and influence signaling cascades [9,19]. There are three general mechanisms for lncRNA regulation: localization and interaction with chromatin, interaction with RNA targets, and protein modulation (Figure 1). A single lncRNA can function via different mechanisms, suggesting that their regulation of gene expression is complex [9]. The ability of lncRNAs to interact with distinct biological molecules has important implications in cancer, and understanding these complex networking interactions may provide novel therapeutic targets [20].

### 2.1. Chromatin Interactions and Transcriptional Regulation

By localizing to chromatin, lncRNAs can regulate pre-transcriptional processes such as gene imprinting, dosage compensation, and epigenetic regulation such as histone modification [19]. For example, the well-characterized lncRNA X-inactive-specific transcript (XIST) localizes to the X chromosome and recruits factors for chromosome inactivation [21]. As X aneuploidy exists in many human cancers, it suggests that dysregulation of X-linked genes is required for malignant transformation [9]. Furthermore, XIST was shown to be upregulated in NSCLC tissues and cell lines and act as an oncogene by recruiting enhancer of zeste homolog 2 (EZH2), a member of the polycomb repressive complex 2 (PRC2), to the promoter of the tumor suppressor Krüppel-like factor 2 (KLF2) to initiate histone methylation to repress KLF2 expression [22]. As KLF2 silencing partly reversed the tumorigenic effects of XIST silencing, it suggests that XIST acts as an oncogene by epigenetically silencing KLF2 by directly binding to EZH2. However, as the pro-oncogenic effects were only partly reversed by KLF2, it suggests that XIST may regulate other possible targets [22].

LncRNAs can regulate transcription by controlling chromatin architecture and recruiting regulatory molecules such as transcription factors to specific sites [9,11]. For example, chromatin looping by lncRNAs induces changes in chromatin structure to promote interactions with key regulatory molecules to activate gene expression, as shown in Figure 1 [11]. *HoxA* distal transcript antisense RNA (HOTTIP) is a lincRNA located in the *HoxA* locus that binds to the transcriptional activator WD repeat-containing protein 5 (WDR5) and the mixed lineage leukemia (MLL) complex. HOTTIP facilitates chromatin looping to recruit the two regulatory factors to the *HoxA* locus which activates gene expression [23]. In NSCLC, HOTTIP expression is significantly higher in malignant tissues than normal adjacent tissues, and it was shown to function as an oncogene by regulating Homeobox Protein A3 (HOXA3 expression to increase cell proliferation and invasion [24]. However, further functional studies are required to determine if chromatin looping is responsible for HOTTIP regulation of HOXA3 in NSCLC.

LncRNAs can also regulate gene expression by acting as scaffolds, to allow the assembly of regulatory complexes for transcriptional activation or repression, depending on the molecules recruited (Figure 1) [11]. The lncRNA antisense noncoding RNA in the *INK4* locus (ANRIL) functions as a scaffold for different chromatin remodeling complexes, such as PRC1, PRC2, and WDR5 [17,25]. In NSCLC, ANRIL was shown to interact with PRC2 to repress the expression of KLF2 and p21. This led to reduced NSCLC cell proliferation and apoptosis, suggesting that scaffold lncRNAs such as ANRIL may be important in NSCLC tumorigenesis [26].

### 2.2. RNA Interactions

LncRNAs are known to function in gene expression through their interaction with different RNA targets [9]. Interactions between mRNAs and lncRNAs can regulate RNA metabolism through the recruitment of factors involved in splicing, mRNA stability, and translation (Figure 1) [9,19]. Metastasis-associated lung adenocarcinoma transcript 1 (MALAT1) is a well-characterized lncRNA that is thought to be involved in the alternative splicing of mRNAs. Indeed, nuclear-retained MALAT1 was shown to interact with and influence the distribution of serine/arginine splicing factors (SRSF), and MALAT1 depletion led to changes in the alternative splicing pattern of a set of pre-mRNAs [27]. MALAT1 is overexpressed in NSCLC and contributes to lung tumorigenesis, as suppression of MALAT1 in A549 NSCLC cells suppressed clonogenic growth and migration [16,28]. MALAT1 was shown to increase SRSF7 levels by inhibiting miR-347b-5p to enhance NSCLC tumorigenesis in vitro [29]. Therefore, MALAT1 may affect the splicing of key tumor suppressor and oncogenic mRNAs to promote tumorigenesis [16,27].

Furthermore, lncRNAs can modulate miRNAs by acting as “miRNA sponges” or competing endogenous RNAs (ceRNAs) that competitively bind to and sequester miRNAs, reducing their regulatory effect on the destined mRNA [Figure 1] [9,10]. Nuclear-enriched abundant transcript 1 (NEAT1), which is highly expressed in NSCLC patients, was shown to function as a ceRNA for miR-377-3p. Direct binding of NEAT1 to miR-377-3p led to de-repression of oncogenic E2F transcription factor (E2F) 3, a direct target of miR-377-3p [30]. In vitro studies showed that NEAT1 promotes NSCLC cell growth by inhibiting the miR-377-3p/E2F3 axis leading to activation of the E2F3 pathway [30,31]. NEAT1 was also shown to “sponge” the miR-181a-5p, upregulating its target gene high-mobility group protein B2 (HMGB2) and increasing NSCLC cell proliferation and invasion [32]. The ability of NEAT1 to interact with distinct miRNAs and modulate NSCLC tumorigenesis via distinct mechanisms suggests a complex regulatory network exists. It is worth noting that, to confirm that NEAT1 acts as a ceRNA, Sun et al. and Li et al. used siRNAs to knockdown NEAT1 and investigate the expression levels of miR-377-3p and miR-181a-5p, respectively [30,32]. However, Zhang et al. failed to check miR-377-3p expression levels, highlighting a limitation for many studies when investigating the function of lncRNAs as “miRNA sponges” [31].

### 2.3. Protein Regulation

LncRNAs can modulate protein functions by promoting the assembly of protein complexes, disrupting protein–protein interactions and affecting their cellular localization [9]. As mentioned above, lncRNAs can modulate protein interactions and the formation of regulatory complexes to regulate all stages of gene expression [23,27,33]. LncRNAs are also able to influence important signaling pathways by modulating proteins directly or indirectly through miRNAs (Figure 1) [9]. In a similar mechanism to NEAT1 modulation of the E2F3 pathway, MALAT1 was shown to directly interact and “sponge” miR-206, leading to activation of the Protein kinase B (AKT)/mammalian target of rapamycin (mTOR) pathway to promote epithelial-to-mesenchymal transition (EMT) and migration of A549 and H1299 NSCLC cells [34].

## 3. LncRNA Dysregulation in NSCLC

Many large-scale analyses investigated differentially expressed lncRNAs in NSCLC [35,36]. As shown in Table 1, most upregulated lncRNAs in NSCLC function as oncogenes, whereas less commonly downregulated lncRNAs function as tumor suppressors (Table 2). However, for some lncRNAs, there is controversy over their role as tumor promoters or suppressors. This review further discusses mechanisms behind well-characterized lncRNAs in NSCLC, as well as the role of novel and less well-established lncRNAs.

### 3.1. Oncogenic LncRNAs

#### 3.1.1. Proliferation and Survival

Dysregulation of the cell cycle is common in cancer to confer a proliferative advantage to cells, where key checkpoint regulators such as cyclins and cyclin dependent kinases (CDKs) are commonly dysregulated [117]. For example, lncRNA *JPX* transcript and XIST activator (JPX) promoted cell-cycle progression and NSCLC cell proliferation by acting as a ceRNA for miR-145-5p to upregulate its mRNA target, cyclin D2 [54]. The lncRNAs XIST, small nucleolar RNA host gene 15 (SNHG15), and *HNF1A* antisense RNA 1 (HNF1A-AS1) led to the upregulation of CDK8, CDK14, and CDK6, respectively, to increase NSCLC growth and proliferation, by acting as ceRNAs to sponge the inhibitory miRNAs miR-16, miR-486, and miR-149-5p, respectively [50,87,93]. Cell-cycle progression proteins can also be regulated by lncRNAs. For example, Cip1 interacting with zinc finger protein 1 (Ciz1) was recently shown to regulate the Gap 1 (G1) to synthesis (S) phase (G1-to-S) transition in cancer and lncRNA differentiation antagonizing nonprotein coding RNA (DANCR) was shown to sponge miR-214-5p, upregulating its target Ciz1 and increasing NSCLC cell proliferation [38,118]. Furthermore, GTPase G1-to-S phase transition 1 (GSPT1) was upregulated by lncRNA *DLX6* antisense gene 1 (DLX6-AS1) due to inhibition of miR-27-3p, resulting in increased NSCLC cell proliferation [40]. In addition, LINC00339 promoted NSCLC proliferation and progression by sponging miR-145 to upregulate the cell-cycle regulator Forkhead box protein 1 (FOXM1) [56]. Therefore, lncRNA regulation of inhibitory miRNAs is important for NSCLC cell-cycle progression and proliferation.

Oncogenic E2F family members, E2F1, E2F2, and E2F3 play important roles in G1-to-S cell-cycle transition [117]. LncRNA HOXA transcript induced by transforming growth factor (TGF)-β (HIT) was shown to interact directly with E2F1 and modulate E2F1 promoter binding to increase expression of its target genes to enhance NSCLC cell proliferation [48]. LINC00461 also led to E2F1 upregulation by sponging miR-4478, where E2F1 could also bind to the promoter of LINC00461 to induce its transcription and initiate a positive feedback loop for NSCLC progression [57]. LncRNA *FLVCR1* antisense gene 1 (FLVCR1-AS1) upregulated E2F3 by acting as a ceRNA for miR-573, promoting NSCLC proliferation and progression [44]. Therefore, lncRNA regulation of the E2F pathway directly or indirectly through miRNAs is also important for NSCLC growth and progression.

As well as upregulating positive cell-cycle regulators, lncRNAs can also epigenetically repress the expression of cell-cycle inhibitors such as cyclin-dependent kinase inhibitor 1A (CDKN1A or p21) and tumor protein 53 (p53) [117]. Three lncRNAs upregulated in NSCLC, *AFAP1* antisense gene 1 (AFAP1-AS1), small nucleolar RNA host gene 20 (SNHG20), and LINC01088, were shown to directly interact with EZH2, increasing binding and methylation of the p21 promoter and silencing of p21 expression [37,61,119]. AFAP1-AS1 knockdown reduced growth in vitro and in vivo and silencing of SNHG20, and LINC01088 reduced NSCLC proliferation and cell-cycle progression [37,61,119]. LncRNAs plasmacytoma variant translocation 1 (PVT1) and prostate cancer-associated transcript 6 (PCAT-6) were also shown to bind to EZH2, and recruit it to the large tumor suppressor kinase 2 (LATS2) promoter, repressing LATS2 transcription [75,78]. PCACT-6 promoted NSCLC growth in vitro and in vivo through epigenetic silencing of LATS2, but further mechanistic insight needs to be discussed [75]. Downregulation of PVT1 or overexpression of LATS2, however, decreased E3 ubiquitin ligase (MDM2) expression and subsequent inhibition of p53. Therefore, lncRNA repression of LATS2 may promote cell proliferation and growth through inhibition of the MDM2–p53 pathway [78]. PVT1 also sponged miR-526b, increasing EZH2 levels and generating a positive feedback mechanism to enhance NSCLC progression [79]. LncRNA lung cancer associated transcript 1 (LUCAT-1) was shown to repress both p21 and p57 expression through interaction with the PRC2 complex to promote NSCLC cell proliferation [67]. Furthermore, LINC01234 acted as a scaffold to bind LSD1 and EZH2 and repress BTG anti-proliferation factor 2 (BTG2) to enhance NSCLC tumorigenesis [62]. Therefore, epigenetic silencing of cell cycle and proliferative inhibitors is a common mechanism for lncRNA regulation of NSCLC proliferation.

In addition to regulation of NSCLC proliferation, lncRNAs are implicated in suppression of apoptosis. For example, lncRNA small nucleolar RNA host gene 6 (SNHG6) competitively sponged miR-490-3p to increase remodeling and splicing factor 1 (RSF1) to promote proliferation and inhibit apoptosis, suggesting the importance of pre-transcriptional complexes in NSCLC cell proliferation and survival [89]. XIST, as well as regulating CDK8, can upregulate the anti-apoptotic factor B-cell lymphoma type 2 (BCL-2) to inhibit apoptosis by sponging miR-449a [93,94]. LncRNAs XLOC-008446 and prostate cancer associated transcript 1 (PCAT-1) can act as ceRNAs for miR-874 and miR-149-5p, respectively, to increase the anti-apoptotic regulators x-linked inhibitor of apoptosis (XIAP) and leucine-rich repeats and immunoglobulin-like domains 2 (LRIG2), respectively [74,97]. Therefore, lncRNA regulation of miRNAs is also important to promote cell survival, as well as proliferation.

#### 3.1.2. Invasion and Migration

Matrix metalloproteinases (MMPs) are key enzymes in tumor dissemination, largely through their role in ECM degradation and proteolytic breakdown of tissue barriers to invasion [120]. In addition to epigenetic silencing of LATS2, PVT1 also acts as a ceRNA for tumor suppressor miRNAs miR-200a and miR-200b to increase their target MMP-9, which promoted the invasive ability of NSCLC cells [78,80]. PVT1 also sponged miR-148 to increase Ras-related protein Rab-34 (RAB34), a GTPase that regulates surface proteins for invasion, to enhance the migration and invasion of NSCLC cells [81]. In a similar mechanism, XLOC-008466 and lncRNA 1308 increase MMP-2 and a disintegrin and metalloproteinase domain 15 (ADAM15) by competitively sponging miR-874 and miR-124, respectively, to increase the invasion of NSCLC cells [65,97]. XLOC-008466 can, therefore, regulate the invasion, as well as apoptosis, of NSCLC cells, through regulation of miR-874 [97]. CeRNA sponging by lncRNAs can also increase transcription factors that regulate MMPs. For example, lncRNA *HOXD* cluster antisense RNA 1 (HOXD-AS1) sponging of miR-133b upregulates its target HOXAD, leading to the upregulation of MMP-9 [53]. SNHG6 was also shown to upregulate MMP-9, as well as MMP-2, by competitively binding miR-994 and miR-181d-5p and increasing ETS proto-oncogene transcription factor 1 (ETS1) activity [90]. MMP-9 expression was also shown to be epigenetically regulated by lncRNA myocardial infarction associated transcript (MIAT), which interacts with the MLL complex to reduce silencing of the MMP-9 promoter [70]. Therefore, various lncRNAs can regulate MMP expression via a variety of mechanisms to promote NSCLC invasion.

XIST was shown to positively regulate paxillin, a focal adhesion protein, by acting as a ceRNA for miR-137 to modulate attachments between the cell and the ECM for NSCLC migration [95]. Due to functions in proliferation, apoptosis, and invasion, XIST may be a promising therapeutic target in NSCLC [93,94,95]. In addition to changes in ECM attachments, cellular gene expression changes are required for NSCLC invasion and migration. LncRNAs SNHG15 and LINC01234, two lncRNAs that also function in cell-cycle regulation, act as ceRNAs to increase oncogenes zinc finger protein 217 (ZNF217) and Vav guanine nucleotide exchange factor 3 (VAV3), respectively, to enhance proliferation and invasion [62,88]. Similar to LINC01234, lncRNA DUXAP8 acts as a scaffold to recruit histone modifiers EZH2 and LSD1 to repress early growth response protein 1 (EGR1) and Rho-related GTP binding protein (RHOB), respectively. EGR1 and RHOB are both tumor suppressors where loss of expression is associated with invasion and metastasis, whereas DUXAP8-induced suppression in NSCLC reduced proliferation and invasion [41]. Therefore, lncRNA regulation of proteins for invasion and migration is important in NSCLC.

#### 3.1.3. EMT and Metastasis

EMT is a key process that allows cancer cells to adopt a more invasive and migratory phenotype for tumor migration, invasion, and metastasis [9] Zinc finger E-box-binding homeobox (ZEB) proteins are key inducers of EMT, and several lncRNAs in NSCLC can regulate their expression and activity. For example, XIST and lncRNA *SOX-2* overlapping transcript (SOX2OT) act as ceRNAs for miR-367/miR-141 and miR-132, respectively, to increase ZEB2 expression and EMT [91,96]. LncRNAs LINC00673, human testis developmental related gene 1 (TDRG1), and *ZEB1* antisense RNA 1 (ZEB1-AS1) increase ZEB1 expression and EMT by sponging mir-150-5p, miR-873-5p, and miR-409-3p, respectively [59,92,98]. ZEB1 can also bind to the promoter of ZEB-AS1 to increase its expression, acting in a positive feedback loop to regulate EMT [98]. This is a similar mechanism to E2F1 feedback on LINC00461 expression to regulate cell-cycle progression and proliferation [57]. HIT can interact directly with ZEB1, increasing its stability and binding to the CDH1 promoter to repress E-cadherin, a marker of EMT [49]. Furthermore, lncRNA *HOXA11* antisense 1 (HOXA11-AS1) can interact with EZH2 and DNMT1 to repress miR-200b expression and increase miR-200b targets ZEB1 and ZEB2 for enhanced EMT [51]. HOXA11-AS1 also sponged miR-148a-3p, increasing DNA methyltransferase 1 (DNMT1) expression and generating a feedback loop for EMT, similar to PVT1 upregulation of EZH2 [52,79]. Twist family BHLH transcription factor 1 (TWIST), another inducer of EMT, was shown to be increased by LINC01296 inhibition of miR-598, and TWIST enhanced the expression of LINC01296 to enhance EMT [64]. Therefore, feedback mechanisms are important in and add to the complexity of lncRNA regulation in NSCLC EMT.

*YES*-associated protein 1 (YAP1), a regulator of the hippo pathway, is another inducer of EMT, and lncRNA nicotinamide nucleotide transhydrogenase-antisense RNA1 (NNT-AS1) led to an increase in YAP1 expression by sponging miR-22-3p, increasing NSCLC migration, invasion, and EMT [73]. *SRY*-box transcription factor 4 (SOX4), a novel epigenetic regulator of EMT, was increased by two lncRNAs DANCR and *LEF1* antisense RNA 1 (LEF1-AS1) by sponging mir-138 and miR-489, respectively, increasing NSCLC migration and metastasis [39,55]. E-cadherin was downregulated by EZH2 induced epigenetic silencing mediated by *FEZF1* antisense RNA 1 (FEZF1-AS1), decreasing cell-to-cell adhesions and increasing EMT [43]. Therefore, lncRNAs can regulate multiple inducers of EMT in NSCLC.

LncRNAs were also shown to regulate proteins associated with metastasis. For example, lncRNA long stress-induced non-coding transcript 5 (LSINCT5) interacted with metastasis-associated transcription factor, high-mobility group AT-hook 2 (HMGA2), protecting it from proteasome-mediated degradation and increasing migration of NSCLC cells [66]. Furthermore, lncRNA *ZNFX1* antisense RNA 1 (ZFAS1) acted as a ceRNA for miR-150 to increase HMGA2, resulting in increased NSCLC invasion [99]. LINC00673 modulated EZH2 epigenetic silencing of HOXA5, a tumor suppressor that inhibited NSCLC metastasis by regulating cytoskeletal remodeling [60]. Metaherin, another oncogene implicated in metastasis in NSCLC, was upregulated by small nucleolar RNA host gene 1 (SNHG1) and prostate cancer non-coding RNA 1 [PRNCR1] by sponging miR-145-4p and miR-126-5p, respectively, increasing EMT, migration, and invasion of NSCLC cells [77,84]. LINC00525 also acts as a ceRNA, for miR-338-3p, increasing NSCLC cell invasion and migration by increasing insulin receptor substrates 2 (IRS2), a signaling adapter protein that is implicated in cancer progression and metastasis [58]. Therefore, lncRNAs can act via multiple mechanisms to regulate EMT and metastasis in NSCLC.

#### 3.1.4. Regulation of Key Oncogenic Signaling Pathways

LncRNAs can also regulate oncogenic signaling pathways that control growth, proliferation, apoptosis, and metastasis by modulating pathway activation and key signaling components [9]. For example, LINC01288 can interact directly with and increase the stability of interleukin 6 (IL-6) mRNA, leading to upstream activation of the signal transducer and activator of transcription 3 (STAT3) pathway, promoting NSCLC proliferation, growth, and invasion [63]. Furthermore, MALAT1 and H19 were shown to act as ceRNAs for miR-124 and miR-17, respectively, to increase their target mRNA STAT3 [46,68]. Therefore, direct upregulation of STAT3 by lncRNAs can also enhance NSCLC cell tumorigenesis.

H19 was also shown to regulate pyruvate dehydrogenase 1 (PDK1) by sponging miR-138 and increasing NSCLC proliferation through activation of the phosphatidylinositol 3-kinase (PI3K)/AKT pathway [47]. Insulin-like growth factor 2 (IGF-2), an upstream activator of the PI3K/AKT pathway, was increased by lncRNAs NEAT1 and GM15290 to enhance NSCLC cell proliferation, invasion, and migration. IGF-2 expression was increased by NEAT1 sponging of the tumor suppressor miR-let7a and lncRNA GM15290 sponging of miR-615-5p [45,72]. In highly metastatic NSCLC SPCA-1scl cells, lncRNA Meta-LNC9 interacted directly with phosphoglycerate kinase 1 (PGK1), protecting it from degradation and leading to activation of the AKT/mTOR pathway to enhance cell migration, invasion, and metastases. Meta-LNC9 also regulated its own transcription by modulating cAMP response element-binding protein (CREB1), generating a positive feedback loop to enhance metastasis [69]. Furthermore, focally amplified lncRNA on chromosome 1 (FAL1) lncRNA increased BMI1 polycomb ring finger oncogene (BMI1) levels, which downregulated phosphatase and tensin homolog (PTEN), an inhibitor of the PI3K/AKT pathway, increasing AKT activation and NSCLC tumorigenesis [42]. Therefore, lncRNAs can also regulate pathway inhibitors to upregulate NSCLC oncogenic signaling.

*PRKCZ* antisense RNA 1 (PRKCZ-AS1) led to enhanced mitogen-activated protein kinase (MAPK) signaling by inhibiting mir-766-5p and increasing its target MAPK1 to enhance NSCLC proliferation and migration [76]. PVT1 can also regulate activation of MAPK signaling by sponging miR-145-5p and increasing integrin subunit beta 8 (ITGB8), as well as activate the wingless (Wnt)/β-catenin pathway by sponging mir-361 and increasing *SRY*-box transcription factor 9 (SOX9) for enhanced NSCLC tumorigenesis [82,83]. As discussed previously, PVT1 was shown to have roles in cell-cycle regulation and the regulation of proteins for invasion and migration and, therefore, be a promising therapeutic target due to its multiple functions in NSCLC progression. SNHG1 also increased SOX9 by sponging miR-101-3p to increase Wnt signaling and NSCLC proliferation and invasion [85]. SNHG1 also could sponge miR-361-3p to increase *FRAT* regulator of WNT signaling Pathway 1 (FRAT1) to enhance NSCLC cell tumorigenesis, suggesting that SNHG1 activation of the Wnt pathway through regulation of distinct lncRNAs is important and complex in NSCLC [86].

### 3.2. Tumor Suppressor LncRNAs

#### 3.2.1. Proliferation and Apoptosis

LncRNA growth arrest specific 5 (GAS5) was shown to act as a tumor suppressor in NSCLC by post-transcriptionally regulating p53, p21, and E2F1 to inhibit tumor growth and increase NSCLC cell apoptosis, but not migration and invasion [105]. GAS5 is downregulated in NSCLC, and low expression correlates with a poor prognosis, in contrast to oncogenic lncRNAs such as HIT and LUCAT-1 that are upregulated in NSCLC and positively regulate proliferation [48,67,105]. Long intergenic non-protein coding RNA, p53-induced transcript (LINC-PINT) acts as a ceRNA of miR-208-3p to upregulate programmed cell death protein 4 (PCDC4), reducing NSCLC cell proliferation and cell-cycle progression [107]. Although the exact mechanism was not defined, lncRNA MIR503-HG also reduced NSCLC proliferation by downregulating cyclin D1 expression [114]. However, MIR503-HG was shown to have a contradictory role as an oncogene by sponging miR-489-3p and mir-625-5p and promoting NSCLC cell proliferation and reduced apoptosis [71]. As a defined mechanism for MIR503-HG’s role in NSCLC is not yet elucidated, further work is required to understand its exact role as a promoter or suppressor of NSCLC.

LncRNAs LINC00702 and *MAGI2* antisense RNA 3 (MAGI2-AS3) can act as ceRNAs for miR-510 and miR-23a-3p, respectively, to increase PTEN expression and reduce NSCLC proliferation [109,111]. Both lncRNAs are downregulated in NSCLC, and their function opposes oncogenic lncRNA FAL1-BMI1 that downregulates PTEN to enhance AKT-induced tumorigenesis [42,109,111]. LncRNA maternally expressed 3 (MEG3) acts as a ceRNA for miR-7-5p to inhibit BCL-2 and promote BCL-2-like factor 4 (BAX), enhancing NSCLC apoptosis [112]. In contrast, XIST acts as a ceRNA to increase BCL-2, and it may be upregulated to reverse the tumor suppressor function of MEG3 [94,112]. Therefore, tumor suppressor lncRNAs can regulate proliferation via similar mechanisms to oncogenic lncRNAs, albeit to suppress NSCLC progression.

#### 3.2.2. EMT and Metastasis

Tumor suppressor lncRNAs can reduce EMT by downregulating key inducers and reducing EMT induction and metastasis. For example, NF-κB (NF-κB)-interacting lncRNA (NKILA) interacts directly with NF-κB to decrease NF-κB inhibitor alpha (IKBa) phosphorylation, inhibiting NF-κB nuclear translocation and zinc finger protein SNAI1 (SNAIL1)-induced EMT, inhibiting NSCLC cell migration and invasion [115]. NKILA could also reduce EMT by inhibiting interleukin 11 (IL-11) mRNA levels, reducing STAT3 phosphorylation and EMT induction [116]. Therefore, NKILA may function transcriptionally and post-transcriptionally to regulate EMT inhibition in NSCLC. LncRNA *BRE* antisense RNA 1 (BRE-AS1) directly interacted with STAT3 to reduce its binding to and derepress the nuclear receptor subfamily 4 group A member 3 (NR4A3) promoter to reduce NSCLC cell growth, survival, and EMT [100]. Furthermore, LINC8150 reduced STAT3 activation by sponging miR-199b-5p and increasing caveolin-1 (CAV1), causing a reduction in NSCLC cell migration and invasion in vitro and EMT and metastases in vivo [110]. Therefore, negative regulation of the STAT3 pathway by tumor suppressor lncRNAs is important for inhibition of NSCLC EMT and metastasis.

LncRNA Fork-head box F1 (FOXF1) antisense RNA 1 (FOXF1-AS1) also inhibited EMT by directing EZH2 repression of FOXF1 expression to inhibit migration and invasion of NSCLC cells [103]. LncRNA *FTX* transcript, XIST regulator (FTX) acted as a ceRNA for miR-200a-3p to increase Fork-head box protein A2 (FOXA2) to inhibit NSCLC cell EMT and metastasis [104]. Interestingly, the miR-200 family is generally associated with tumor suppressors and oncogenic lncRNAs such as PVT1 silence miR-200a and miR-200b to enhance NSCLC tumorigenesis [80]. As FTX also inhibits miR-200 family member miR-200a-3p to suppress NSCLC progression, it suggests that miRNA regulation by lncRNAs is complex in NSCLC [104]. *GAS5* antisense RNA 1 (GAS5-AS1) also inhibited NSCLC cell EMT to inhibit migration and invasion of NSCLC cells [106]. As GAS5 sense lncRNA inhibited NSCLC cell proliferation but not migration and invasion, transcription of lncRNAs in both the sense and antisense direction may be important for inhibition of NSCLC tumorigenesis [105,106].

In addition to a role in inhibiting apoptosis, MEG3 can increase NSCLC migration, invasion, and metastasis by sponging miR-650 and increase metastatic suppressor solute carrier family 34 member 2 (SLC34A2) [112,113]. LncRNA DiGeorge syndrome critical region gene 5 (DGCR5) also acts as a ceRNA to sponge miR-211-5p to increase EPH receptor B6 (EPHB6) to reduce NSCLC cell growth, invasion, and migration [101]. Furthermore, LINC00261 acted as a ceRNA for miR-552-3p to increase Wnt pathway suppressor secreted Frizzled-related protein 2 (SRFP2) to increase NSCLC cell apoptosis and inhibit invasion [108]. In contrast to oncogenic lncRNAs that upregulate MMPs to enhance invasion, lncRNA *FOXF1* adjacent non-coding developmental regulatory RNA (FENDRR) upregulates tissue inhibitors of MMP-2 (TIMP2) by sponging mir-176 to suppress NSCLC migration, invasion, and metastasis [102]. Tumor suppressor lncRNAs can, therefore, suppress NSCLC EMT and metastasis via distinct mechanisms.

## 4. Clinical Potential of LncRNAs in NSCLC

### 4.1. Diagnostic and Prognostic Potential

With a five-year survival of around 15%, NSCLC is associated with a poor prognosis, which is largely attributable to poor detection and late diagnosis [121]. Current traditional detection methods typically have low sensitivity and specificity, limiting early detection, and novel biomarkers are required for improved molecular diagnosis and prognosis [122]. Although proteins are commonly used as markers for diagnosis, lncRNAs are advantageous as clinical indicators as they are stable and highly tissue-specific, and they can be detected in various bodily fluids [122,123]. Compared to traditional biopsies, lncRNA-based biomarkers may also be better endured by patients and minimally invasive [123]. As reviewed in Shi et al., lncRNAs could be used to distinguish early-stage disease from healthy patients at a high sensitivity and specificity, as well as providing prognostic insight into the risk of metastases and recurrence [124]. For example, prostate cancer antigen 3 (PCA3) is an overexpressed lncRNA in prostate cancer (PCa) that contributes to PCa progression by modulating androgen signaling, and PCA3 urine levels were successfully used as a biomarker for PCa diagnosis [125].

In NSCLC, MALAT1 was shown to function in many aspects of NSCLC tumorigenesis, and its potential as a biomarker detected in fluids was studied. MALAT1 detection in NSCLC blood samples was shown to have a high specificity compared to cancer-free controls, as well as minimal invasiveness, suggesting its promise as a diagnostic tool. However, with a low sensitivity of 56%, it suggests that the use of MALAT1 as a single diagnostic biomarker may not be feasible in NSCLC [126]. XIST and HIF1A-AS1 both play important roles in NSCLC and are upregulated in NSCLC tissues and serums. Interestingly, the combination of XIST and HIF1A-AS1 yielded a higher positive diagnostic rate compared to either lncRNA alone [127]. Therefore, the use of multiple lncRNAs may improve the sensitivity and diagnostic efficiency of NSCLC; however, a commonly recognized group of lncRNAs is required. Li et al. performed a meta-analysis investigating the prognostic potential of MALAT1 and, although MALAT1 sensitivity was low for individual diagnostic testing, they found that MALAT1 could be used significantly as an independent prognostic factor for overall survival in NSCLC [128]. Zhang et al. investigated the clinical potential of multiple lncRNAs in NSCLC patients and found that a poor prognosis was associated with high expression of H19, MALAT1, and *Hox* antisense intergenic RNA (HOTAIR) and low expression of taurine upregulated gene 1 (TUG1) and p21-associated ncRNA DNA damage activated (PANDA) [129]. Therefore, individual lncRNAs, as well as combinations, may be important in investigating the prognosis of NSCLC.

### 4.2. Therapeutic Potential

The main form of treatment for NSCLC is curative intent surgery; however, with patients typically presenting at late-stage disease, surgery has limited efficacy, and platinum-based chemotherapy is the standard of care [121]. Furthermore, with increasing resistance to chemotherapies and targeted treatments, novel approaches need to be explored [122]. Due to the function of lncRNAs in all aspects of NSCLC tumorigenesis and the regulation of key signaling pathways, lncRNAs may be promising therapeutic targets. Furthermore, many lncRNAs are associated with enhanced chemoresistance and, therefore, their targeting may also restore cancer cell sensitivity to chemotherapeutic drugs [122].

There are various different approaches for targeting lncRNAs in cancer including RNA interference (RNAi)-based gene silencing, antisense oligonucleotide (ASO)-based treatment, small-molecule modulators of lncRNA–protein interactions, and the delivery of tumor suppressor lncRNAs [122,130]. For example, siRNA interference of HOTAIR decreased the migration and invasion of NSCLC cells in vitro and reduced metastases in a HOTAIR siRNA xenograft mouse model [131]. Furthermore, HOTAIR siRNA-mediated knockdown also increased the sensitivity of NSCLC cells to cisplatin treatment (Table 3) [132]. However, RNAi can have off-target effects and can be problematic for nuclear RNAs, with many lncRNAs functioning in the nucleus in NSCLC. In contrast, ASOs are advantageous due to their high affinity and reduced toxicity due to relatively low off-target effects [133]. In a MALAT1 knockout mouse model, mice injected with MALAT1 ASOs had reduced lung tumor nodules and volume compared to untreated mice (Table 3). Therefore, ASO inhibition of MALAT1 inhibited NSCLC metastasis and may provide a promising therapeutic approach in NSCLC [134]. Many lncRNAs function in NSCLC by interacting with or regulating the epigenetic regulator EZH2 (Table 1), to promote NSCLC tumorigenesis. High-throughput screening methods to identify small-molecule inhibitors of the lncRNA–EZH2 interaction were developed. which may lead to the identification of general inhibitors targeting the RNA-binding pocket, as well as specific inhibitors targeting particular lncRNA interactions [135]. Targeting lncRNA interactions with proteins, such as EZH2, may be a strategy to reduce off-target effects and increase targeted specificity (Table 3). MEG3 was shown to be downregulated in NSCLC, and it inhibits NSCLC cell apoptosis, invasion, migration, and metastasis, as well as increasing sensitivity to cisplatin [112,113,136]. Overexpression of MEG3 inhibited tumorigenesis in vivo and reduced NSCLC cell proliferation, as well as induced apoptosis in vitro, suggesting that delivery of tumor suppressor lncRNAs such as MEG3 may potentially be an alternative therapeutic option in NSCLC (Table 3) [137]. However, further work is required to assess the effectiveness of tumor suppressor lncRNA delivery as a therapeutic option in the clinic.

### 4.3. Limitations and Future Prospectives

Despite evidence supporting the role of lncRNAs as therapeutic targets in NSCLC, there are limitations to targeting and studying the effectiveness of lncRNAs. For example, the lack of a protein product restricts treatment approaches largely to nucleic acid-based therapies [138]. Although successful, their limitations include off-target effects, difficulty crossing the cellular plasma membrane, and reduced bioavailability [139]. Furthermore, unlike proteins, the three-dimensional (3D) structure of lncRNAs remains largely unknown, and a lack of conserved domains could hinder the design of small-molecule inhibitors. A lack of full understanding of the mechanisms and regulatory networks for many lncRNAs in NSCLC may also limit specific targeting strategies to reduce toxicities [138]. Although some lncRNAs in NSCLC are conserved, such as MALAT1, NEAT1, and H19, many lack conservation across species, which can impede investigations and pre-clinical studies in animal models [138,139]. Furthermore, the lungs are a difficult site for siRNA and ASO administration in vivo, which may hinder pre-clinical studies for lncRNA therapies in NSCLC. There is difficulty in initial lung delivery due to physical barriers such as mucosa and cilia, as well as avoiding the immune system. Furthermore, entry past the target cell membrane into the correct subcellular compartment and escaping endosome destruction is difficult to achieve [140]. Further work is, therefore, required to improve drug delivery to the lungs in vivo to accurately study the effectiveness of lncRNA-based therapies in NSCLC.

The use of ASOs in the treatment of neurodegenerative disorders showed promise, with the FDA approval in 2016 of Nusinersen and Eteplirsen for the treatment of spinal muscular atrophy and Duchenne muscular dystrophy, respectively [141,142]. The success of ASO-based therapies paved the way for investigating lncRNA-driven ASO treatments in cancer and, despite the limitations of lncRNAs as therapeutic targets, there is hope that they can become a reality in the future for NSCLC.

## 5. Conclusions

LncRNAs were shown to function in many important cellular processes, with their role in cancer becoming increasingly more apparent [9]. The localization of lncRNAs largely reflects their function, and they can interact with chromatin, protein, and RNA to regulate all stages of gene expression and influence important signaling cascades [9]. Although normally tightly regulated, large-scale analyses revealed that lncRNAs are commonly dysregulated in NSCLC [36]. Many lncRNAs are upregulated and function as oncogenes to enhance NSCLC proliferation, survival, invasion, migration, EMT, and metastasis. Some lncRNAs, such as MALAT1, PVT1, and XIST, can function in multiple aspects of NSCLC tumorigenesis through a variety of different mechanisms. Although the most common mechanism of lncRNA regulation in NSCLC appears to be miRNA sponging, many lncRNAs can also modulate gene expression and protein interactions and stability, suggesting that lncRNA regulation in NSCLC is complex. Tumor suppressor lncRNAs that are downregulated in NSCLC act via similar mechanisms to oncogenes, albeit to suppress cancer progression. As many of the pathways and genes regulated by suppressor lncRNAs are also regulated by oncogenic lncRNAs, oncogenic lncRNAs may be upregulated to reverse the effects of suppressor lncRNAs. Although the mechanisms of action are detailed for the majority of lncRNAs within this review, the mechanisms of action for many lncRNAs are not fully elucidated and are required to ascertain their function as tumor promoters or suppressors.

As the poor prognosis for NSCLC is largely attributable to late diagnosis and the lack of efficient treatments for late-stage disease, novel approaches are required for the management of NSCLC [121]. Combinations of multiple lncRNAs show promise as diagnostic biomarkers, as well as individual or multiple lncRNAs as prognostic indicators. However, further study is required to bring lncRNAs to the clinic as diagnostic tools in NSCLC. Recently, ASOs were successful in the treatment of neurodegenerative diseases, and they may pave the way for lncRNA-based therapies in NSCLC [141,142]. Despite the limitations in studying and designing treatments for lncRNAs, having a deeper understanding of their mechanisms may direct novel treatment approaches to reduce the burden of NSCLC.

## Figures and Tables

**Figure 1 ncrna-06-00025-f001:**
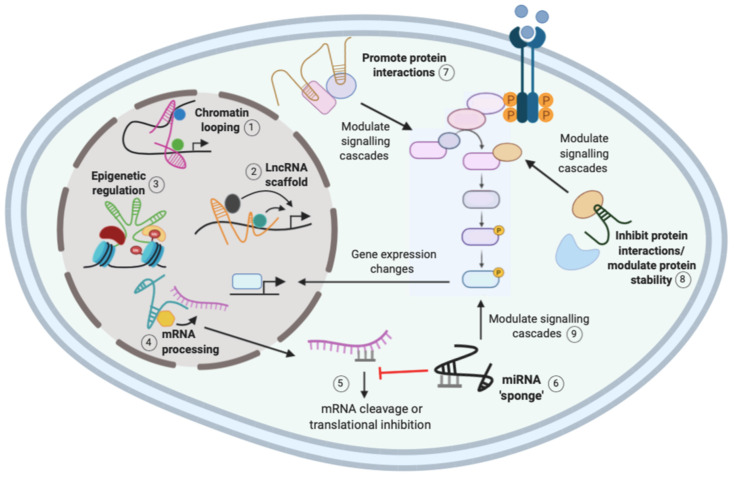
Mechanisms of action of long non-coding RNAs (lncRNAs). The function of lncRNAs is largely reflected by subcellular localization, and their mechanisms include interaction with chromatin, proteins, and RNA. Nuclear lncRNAs can induce chromatin looping (1) or act as a scaffold (2) to recruit multiple regulatory molecules to a gene promoter to activate or repress gene expression. Nuclear lncRNAs can recruit epigenetic regulatory complexes to gene promoters to induce methylation changes to regulate gene expression (3). Nuclear lncRNAs can also recruit regulatory molecules to mRNAs to regulate messenger RNA (mRNA) processing (4). Upon nuclear export for translation, mRNAs can be bound by miRNAs that promote mRNA degradation or inhibition of translation to inhibit mRNA function (5). Cytoplasmic lncRNAs can act as miRNA sponges to competitively bind the miRNAs and release the inhibition on the mRNA (6). Cytoplasmic lncRNAs also modulate protein interactions (7) and stability (8) to regulate signaling cascades and subsequent changes in gene expression. Micro RNA (miRNA) sponging can also modulate signaling cascades by regulating the activity of mRNAs (9). Created with Biorender.com.

**Table 1 ncrna-06-00025-t001:** Oncogenic lncRNAs upregulated in non-small-cell lung cancer (NSCLC).

LncRNA	Mechanism	Function in NSCLC	Ref.
AFAP1-AS1	EZH2 silencing of p21 promoter	Proliferation/growth	[37]
ANRIL	EZH2 silencing of KLF2 and p21	Proliferation/apoptosis	[26]
DANCR	DANCR/miR-214-5p/Ciz1	G1-to-S transition/proliferation	[38]
	DANCR/miR-138/SOX4	EMT/metastasis	[39]
DLX6-AS1	DLX6-AS1/miR-27-3p/GSPT1	G1-to-S transition/proliferation	[40]
DUXAP8	Epigenetically repress EGR1 and RHOB	Proliferation/invasion	[41]
FAL1	FAL1/BMI1/PTEN/PI3K-AKT	Tumorigenesis	[42]
FEZF1-AS1	EZH2 silencing of E-cadherin	EMT	[43]
FLVCR1-AS1	FLVCR1-AS1/miR-573/E2F3	Proliferation	[44]
GM15290	GM15290/miR-615-5p/IGF-2/PI3K-AKT	Tumorigenesis	[45]
H19	H19/miR-17/STAT3	Tumorigenesis	[46]
	H19/miR-138/PDK1/PI3K-AKT	Proliferation	[47]
HIT	Modulate E2F1 activity	Proliferation	[48]
	Increase ZEB1 stability, repress CDH1	EMT	[49]
HNF1A-AS1	HNF1A-AS1/miR-149-5p/CDK6	Growth/proliferation	[50]
HOTTIP	Regulation of HOXA3	Proliferation/invasion	[24]
HOXA11-AS1	MiR-200b repression, ZEB1/ZEB2	EMT	[51]
	MiR-148a-3p/DNMT1 (PF)	EMT	[52]
HOXAD-AS1	HOXAD-AS1/miR-133b/HOXAD/MMP9	Invasion	[53]
JPX	JPX/miR-145-5p/cyclin D2	Cell cycle/proliferation	[54]
LEF1-AS1	LEF1-AS1/miR-489/SOX4	Migration/metastasis	[55]
LINC00339	LINC00339/miR-245/FOXM1	Proliferation	[56]
LINC00461	MiR-4478/E2F1/LINC00461 (PF)	Proliferation	[57]
LINC00525	LINC00525/miR-338-3p/IRS2	Invasion/migration	[58]
LINC00673	LINC00673/miR-150-5p/ZEB1	EMT	[59]
	EZH2 repression of HOXA5	Metastasis	[60]
LINC01088	EZH2 silencing of p21	Proliferation/cell cycle	[61]
LINC01234	Scaffold for LSD1/EZH2, repress BTG2	Proliferation	[62]
	LINC01234/miR-340-5p/27b-3p/VAV3	Proliferation/invasion	[62]
LINC01288	IL-6 stability, STAT3 activation	Tumorigenesis	[63]
LINC01296	MiR-598/TWIST/LINC01296 (PF)	EMT	[64]
LncRNA 1308	LINC1308/miR-124/ADAM15	Invasion	[65]
LSINCT5	HMGA2 stability	Migration	[66]
LUCAT-1	PRC2 repression of p21 and p53	Proliferation	[67]
MALAT1	MALAT1/miR-347b-5p/SRSF7	Splicing	[29]
	MALAT1/miR-206/AKT/mTOR pathway	EMT	[34]
	MALAT1/miR-124/STAT3	Tumorigenesis	[68]
Meta-LNC9	PGK1 stability, AKT/mTOR activation	Invasion/metastasis	[69]
	Meta-LNC9/CREB1/Meta-LNC9 (PF)	Metastasis	[69]
MIAT	Inhibit MLL silencing of MMP-9	Invasion	[70]
MIR-503-HG	MIR-503-HG/miR-489-3p/miR-625-5p	Proliferation/apoptosis	[71]
NEAT1	NEAT1/miR-377-3p/E2F3	Proliferation	[30]
	NEAT1/iR-181a-5p/HMGB2	Proliferation/invasion	[32]
	NEAT1/miR-let7a/IGF-2/PI3K-AKT	Tumorigenesis	[72]
NNT-AS1	NNT-AS1/miR-22-3p/YAP1	Migration/invasion/EMT	[73]
PCAT-1	PCAT-1/miR-149-5p/LRIG2	Invasion	[74]
PCAT-6	EZH2 silencing of LATS2	Proliferation	[75]
PRKCZ-AS1	PRKCZ-AS1/miR-766-5p/MAPK1	Proliferation/migration	[76]
PRNCR1	PRNCR1/miR-126-5p/Metaherin	EMT/migration/invasion	[77]
PVT1	PVT1/EZH2/LATS2/MDM2/p53	Cell proliferation/growth	[78]
	PVT1/miR-526b/EZH2 (PF)	Tumorigenesis	[79]
	PVT1/miR200a/miR-200b/MMP9	Invasion	[80]
	PVT1/miR-148/RAB34	Invasion/migration	[81]
	PVT1/miR-145-5p/ITGB8/MAPK	Tumorigenesis	[82]
	PVT1/miR-361/SOX9/Wnt/β-catenin	Tumorigenesis	[83]
SNHG1	SNHG1/miR-145-4p/Metaherin	EMT/migration/invasion	[84]
	SNHG1/miR-101-3p/SOX9/Wnt/β-catenin	Proliferation/invasion	[85]
	SNHG1/miR-361-3p/FRAT1/Wnt/β-catenin	Tumorigenesis	[86]
SNHG15	SNHG15/miR-486/CDK14	Growth/proliferation	[87]
	SNHG15/miR-211-3p/ZNF217	Proliferation/invasion	[88]
SNHG20	EZH2 silencing of p21	Proliferation/cell cycle	[37]
SNHG6	SNHG6/miR-490-3p/RSF1	Proliferation/inhibit apoptosis	[89]
	SNHG6/miR-994/181d-5p/ETS1/MMP9/2	Invasion	[90]
SOX2OT	SOX2OT/miR-132/ZEB2	EMT	[91]
TDRG1	TDRG1/miR-873-5p/ZEB1	EMT	[92]
XIST	EZH2 silencing of KLF2	Tumorigenesis	[22]
	XIST/miR-16/CDK8	Cell cycle progression	[93]
	XIST/miR-449a/BCL-2	Apoptosis inhibition	[94]
	XIST/miR-137/paxillin	Invasion	[95]
	XIST/miR367/miR-141/ZEB2	EMT induction	[96]
XLOC-008466	XLOC-008466/miR-874/XIAP	Inhibit apoptosis	[97]
	XLOC-008466/miR-874/MMP2	Invasion	[97]
ZEB1-AS1	ZEB1-AS1/miR-409-3p/ZEB1/ZEB1-AS1 (PF)	EMT	[98]
ZFAS1	ZFAS1/miR-150/HMGA2	Invasion	[99]

AFAP-AS1: AFAP antisense gene 1; EZH2: Enhancer of zeste homolog 2; p21: Cyclin-dependent kinase inhibitor 1A; ANRIL: Antisense noncoding RNA in the INK4 locus; KLF2: Krüppel-like factor 2; DANCR: Differentiation antagonizing nonprotein coding RNA; Ciz1: Cip1-interacting zinc finger protein 1; G1-to-S: Gap 1 to synthesis; SOX4: SRY-box transcription factor; EMT: epithelial to mesenchymal transition; DLX6-AS1: *DLX6* antisense gene 1; GSPT1: G1-to-S phase transition 1; EGR1: Early growth response protein 1; RHOB: Rho-related GTP-binding protein; FAL1: Focally amplified lncRNA on chromosome 1; BMI1: BMI1 polycomb ring finger oncogene; PTEN: Phosphatase and tensin homolog; PI3K/AKT: Phosphatidylinositol 3-kinase/Protein kinase B; FEZF1-AS1: *FEZF1* antisense RNA 1; FLVCR1-AS1: LncRNA *FLVCR1* antisense gene 1; E2F3: E2F transcription factor 3; IGF-2: Insulin-like growth factor 2; STAT3: Signal transducer and activator of transcription 3; PDK1: Pyruvate dehydrogenase 1; HIT: *HOXA* transcript induced by transforming growth factor (TGF)-β E2F1: E2F transcription factor 1; ZEB1: Zinc finger E-box-binding homeobox 1; CDH1: E-cadherin; HNF1A-AS1: *HNF1A* antisense RNA 1; CDK6: cyclin dependent kinase; HOTTIP: *HoxA* distal transcript antisense RNA; HOXA3: Homeobox Protein A3; HOXA11-AS1: *HOXA11* antisense 1; ZEB2: Zinc finger E-box-binding homeobox 2; DNMT1: DNA methyltransferase 1; MMP9: Matrix metalloproteinase 9; JPX: *JPX* transcript and XIST activator; LEF1-AS1: *LEF1* antisense RNA 1; FOXM1: Fork-head box protein 1; PF: positive feedback loop; IRS2: Insulin receptor substrates 2; LSD1: Lysine-specific histone demethylase 1A; BTG2: BTG anti-proliferation factor 2; VAV3: Vav guanine nucleotide exchange factor 3; IL-6: Interleukin 6; TWIST: Twist family BHLH transcription factor 1; ADAM15: A disintegrin and metalloproteinase domain 15; LSINCT5: Long stress-induced non-coding transcript 5; HMGA2: High-mobility group AT-hook 2; LUCAT-1: Lung cancer-associated transcript 1; PRC2: Polycomb repressive complex 2; P53: Tumor protein 53; MALAT1: Metastasis-associated lung adenocarcinoma transcript 1; SRSF: Serine/arginine splicing factors; mTOR: Mammalian target of rapamycin; PGK1: Phosphoglycerate kinase 1; NEAT1: Nuclear-enriched abundant transcript 1; HMGB2: High-mobility group protein B2; NNT-AS1: Nicotinamide nucleotide transhydrogenase-antisense RNA1; YAP1: *YES-*associated protein 1; PCAT-1: Prostate cancer-associated transcript 1; LRIG2: Leucine-rich repeats and immunoglobulin-like domains 2; PACT-6: Prostate cancer-associated transcript 6; PRKCZ-AS1: *PRKCZ* antisense RNA 1; MAPK: Mitogen-activated protein kinase; PRNCR1: Prostate cancer non-coding RNA 1; PVT1: Plasmacytoma variant translocation 1; RAB34: Ras-related protein rab-34; ITGB8: Integrin subunit beta 8; SOX9: SRY-box transcription factor 9; SNHG1: Small nucleolar RNA host gene 1; FRAT1: FRAT regulator of WNT signaling Pathway 1; SNHG15: Small nucleolar RNA host gene 15; CDK14: Cyclin-dependent kinase 14; ZNF217: Zinc finger protein 217; SNHG20: Small nucleolar RNA host gene 20; SNHG6: Small nucleolar RNA host gene 6; RSF1: Remodeling and splicing factor 1; ETS1: ETS proto-oncogene transcription factor 1; MMP2: Matrix metalloproteinase 2; SOX2OT: *SOX-2* overlapping transcript; TDRG1: Human testis developmental related gene 1; XIST: X-inactive-specific transcript; CDK8: Cyclin-dependent kinase 8; XIAP: X-linked inhibitor of apoptosis; ZEB1-AS1: *ZEB1* antisense RNA 1; ZFAS1: *ZNFX1* antisense RNA 1. Tumorigenesis includes proliferation, migration and invasion.

**Table 2 ncrna-06-00025-t002:** Tumor suppressor lncRNAs downregulated in NSCLC.

LncRNA	Mechanism	Function in NSCLC	Ref.
BRE-AS1	Inhibit STAT3 at NR4A3 promoter	Growth/survival	[100]
DGCR5	DGCR5/miR-211-5p/EPHB6	Growth/invasion/migration	[101]
FENDRR	FENDRR/miR-176/TIMP2	Migration/invasion/metastasis	[102]
FOXF1-AS1	EZH2 repression of FOXF1	Migration/invasion	[103]
FTX	FTX/miR-200a-3p/FOXA2	EMT/metastasis	[104]
GAS5	Regulation of p21, p53 and E2F1	Inhibit growth/promote apoptosis	[105]
GAS5-AS1	N/A	Migration/invasion	[106]
LINC-PINT	LINC-PINT/miR-208-3p/PCDC4	Proliferation/cell cycle progression	[107]
LINC00261	LINC00261/miR-552-3p/SRFP2	Apoptosis/invasion	[108]
LINC00702	LINC00702/miR-510/PTEN	Proliferation	[109]
LINC8150	LINC8150/miR-199b-5p/CAV1/STAT3	Migration/invasion/EMT/metastasis	[110]
MAGI2-AS3	MAGI2-AS3/miR-23a-3p/PTEN	Proliferation	[111]
MEG3	MEG3/miR-7-5p/BCL-2/BAX	Apoptosis	[112]
	MEG3/miR-650/SLC34A2	Migration/invasion/metastasis	[113]
MIR503-HG	Downregulation of cyclin D1	Reduce proliferation	[114]
NKILA	NKILA/NFKb/IKBa/SNAIL	Migration/invasion/EMT	[115]
	NKILA/IL-11/STAT3	EMT	[116]

BRE-AS1: *BRE* antisense RNA 1; STAT3: Signal transducer and activator of transcription 3; NR4A3: Nuclear receptor subfamily 4 group A member 3; DGCR5: DiGeorge syndrome critical region gene 5; EPHB6: EPH receptor B6; FENDRR: FOXF1 adjacent non-coding developmental regulatory RNA; TIMP2: Tissue inhibitors of MMP 2; FOXF1-AS1: Fork-head box F1 (FOXF1) antisense RNA 1; FOXF1: Fork-head box F1; EMT: epithelial to mesenchymal transition; FTX: FTX transcript, XIST regulator; FOXA2: Fork-head box protein A2; GAS5: Growth arrest specific 5; p21: Cyclin-dependent kinase inhibitor 1A; p53: Tumor protein 53; E2F1: E2F transcription factor 1; GAS5-AS1: GAS5 antisense RNA 1; N/A: non-applicable; LINC-PINT: Long intergenic non-protein-coding RNA, p53-induced transcript; PCDC4: Programmed cell death protein 4; SRFP2: Secreted Frizzled-related protein 2; PTEN: Phosphatase and tensin homolog; CAV1: Caveolin-1; STAT3: Signal transducer and activator of transcription 3; MAGI2-AS3: *MAGI2* antisense RNA 3; MEG3: Maternally expressed 3; SLC34A2: Solute carrier family 34 member 2; NKILA: NF-κB-interacting lncRNA; IKBa: NF-κB inhibitor alpha; SNAIL: Zinc finger protein SNAI1.

**Table 3 ncrna-06-00025-t003:** Therapeutic targeting approaches in NSCLC.

LncRNA	Targeting Approach	Effect in NSCLC	Ref.
HOTAIR	siRNA	Reduced migration and invasion in vitro and reduced metastases in vivo.	[131]
	siRNA	Increased sensitivity to cisplatin.	[132]
MALAT1	ASO	Reduced tumor burden and metastases in vivo.	[134]
lncRNA-EZH2	Small molecule inhibitors targeting lncRNA-protein interactions.	Reduce NSCLC tumorigenesis and off-target effects.	[135]
MEG3	Overexpression of tumour suppressor lncRNA.	Inhibited tumorigenesis in vivo and reduced cell proliferation and apoptosis in vitro.	[137]

HOTAIR: *Hox* antisense intergenic RNA; siRNA: small interfering RNA; MALAT1: Metastasis-associated lung adenocarcinoma transcript 1; ASO: antisense oligonucleotide; lncRNA-EZH2: long non-coding RNA-Enhancer of zeste homolog 2; MEG3: Maternally expressed 3.

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
