# Peer review of "LncRNAs in Non-Small-Cell Lung Cancer"

_ncrna, 2020, doi:10.3390/ncrna6030025_

Round 1

Reviewer 1 Report

The review by L. Ginn and colleagues, it is an overview of lncRNAs role and mechanism of action in NSCLC tumorigenesis. In addition, it describes lncRNAs as novel targets for therapeutic application into the clinic. This review is beautifully written and timely. The information given might be considered as a guide to researchers interested in exploring the promising field of lncRNAs not only in NSCLC but also in other types of malignancies.

I only have a few minor comments:

1) the authors could mention also PMID: 30018188 that provides an ample overview of the  biogenesis, function and interaction of lncRNAs.

2) The authors could add another table that describes the paragraph 4.2. So, the reader can easily identify the lncRNAs and the targeting approaches.

Reviewer 2 Report

Ginn and colleagues present a thorough review of lncRNAs in NSCLC. Overall the paper is complete and well presented. My main feedback is that the authors rather too faithfully repeat the overused cliches of "Once thought be junk DNA..." . I think the argument of junk DNA was discarded about 15 years ago, while the debate about noisy RNA is still in progress and not touched on in this review. The authors also rather too obediently cite papers about sponges and HOTAIR where there remain serious denbates about their utility. Such debates can at least be acknowledged.

L39 "the function of non-coding RNAs (ncRNA) has changed from ‘junk’ transcriptional products to key regulators of a wide range of cellular processes (5). " - their function has not changed. The perception of their function has perhaps changed. But there are still well grounded arguments that many lncRNAs are noise.

L57 "The expression of lncRNAs is regulated at the transcriptional and epigenetic level and tightly controlled." - citation required for this "epigenetic" regulation and "tight" control.

L78 All these statements in this paragraph would be best made with an example and a citation.

L117 HOTAIR has numerous question marks over its functionality (or lack of) and the accuracy of the mentioned mechanistic model (with PRC2). Also, there are hundreds of far better characterised cancer lncRNAs published over the last decade. Therefore, I'm surprised the authors cannot find a better example to illustrate their point than HOTAIR.

L137 Hundreds of papers have been published claiming this or that lncRNA "sponges" microRNAs. However there is major skepticism about this sponge model. A good experimental test is to knock down the lncRNA and check microRNA levels - an experiment that most papers fail to do. Therefore, do the cited papers meet this criteria?
